# Red Mud-Amended Soil as Highly Adsorptive Hybrid-Fill Materials for Controlling Heavy Metal Sewage Seepage in Industrial Zone

**DOI:** 10.3390/ijerph192215043

**Published:** 2022-11-15

**Authors:** Haomin Lei, Xin Xu, Xiaofeng Liu, Junboum Park, Zhongyu Yu, Hao Liu

**Affiliations:** 1College of Construction Engineering, Jilin University, Changchun 130000, China; 2College of Civil Engineering, Taiyuan University of Technology, Taiyuan 030024, China; 3Department of Civil and Environmental Engineering, Seoul National University, Seoul 08826, Republic of Korea

**Keywords:** red mud, industrial sewage, hybrid-fill materials, adsorption capacity, engineering characteristics

## Abstract

With the rapid development of urbanization, the problem of environmental pollution is becoming more and more serious. As a major pollutant, heavy metals have caused serious contamination in soil and groundwater. In order to prevent the diffusion of heavy metals in the soil from industrial sewage, the concept of hybrid-fill layer construction improved by red mud was proposed in this study. This study examines the adsorption capacities of lead and zinc ions and engineering characteristics on red mud-amended soils by direct shear, permeability, adsorption, desorption batch and column tests. Two mixing methods, full particle size displacement mixing and partial particle size displacement mixing, were adopted. The results showed that red mud effectively increased the adsorption capacity of soil to heavy metal ions, and the desorption rate of ions after adsorption was less than 3%, which had good anti-desorption ability. The optimum content of red mud in hybrid-fill material can be determined as 20%. The direct shear test showed that the internal friction angle of hybrid-fill material was 38.9°, and the cohesive force was 30.3 kPa, which met the engineering strength requirements of foundation materials. Based on the test results, red mud can be used as a barrier material to prevent heavy metal contamination in industrial sewage from diffusion, which controls not only heavy metal contamination but also consumes industrial by-products.

## 1. Introduction

The environmental problems caused by industrial sewage are becoming increasingly serious, especially the problem of heavy metal contamination in soil caused by industrial sewage seepage. The heavy metals mainly include chromium (Cr), copper (Cu), zinc (Zn), arsenic (As), cadmium (Cd), nickel (Ni), lead (Pb) and so on. Heavy metal contamination can cause serious damage to the environment and lead to food production reduction, water pollution and other problems [1]. In addition, due to the biological magnification, some heavy metals migrate to the food chain, which seriously threatens human healthy [2,3]. Heavy metals can cause pathological changes in various human tissues after entering the human body. For example, Cd can induce carcinogenesis and a series of cardio-cerebrovascular diseases, As can cause carcinogenesis of cells, and Hg and Pb can cause great damage to the nervous system [4,5,6,7].

When the heavy metal dust dissolves in water, and the industrial sewage due to sewage pipes leak or direct leakage seeps into the industrial complex foundation soil, it will cause serious heavy metal contamination. The type of heavy metal contamination in different areas of China is different, such as heavy metal contamination in mines, heavy metal contamination caused by electronic industry in the east and south, and heavy metal contamination caused by heavy industry in the north [8,9,10].

Heavy metal contamination caused by surface industrial sewage in the industrial zone is generally controlled by preventing the diffusion of heavy metals in foundation soil. Currently, physical and chemical remediation are the two main methods for heavy metal contamination in soil [11]. The physical remediation mainly includes soil washing, soil replacement, soil cover, soil isolation, encapsulation, nano-remediation, etc. [12,13,14]. These methods can effectively prevent the continuous diffusion of heavy metals, but their cost is high for large contamination, and the engineering quantities are large. The chemical remediation mainly includes vitrification technique, electrochemistry, chemical leaching, chemical fixation, adsorption, etc. [15,16,17]. The chemical fixation and adsorption methods are widely used to treat heavy metal contamination in industrial sewage, groundwater and soil.

Red mud is a kind of industrial waste produced by the Bayer Process, which is widely used to produce alumina from bauxite, and it is mainly composed of SiO_2_, CaO, Al_2_O_3_ and other oxides [18]. China is a large alumina-producing country. In 2018, the annual output of red mud in China was about 105 million tons, and about 480~870 million tons of red mud had not been treated harmlessly. Most of them are treated by damming and storing, which occupies land, waste resources, and easily causes environmental pollution [19]. At present, the utilization rate of red mud in China is less than 4%, so the comprehensive utilization of red mud has always been hot [20]. Red mud has a porous structure, large porosity, small specific surface area and large cation exchange capacity, so it has a strong adsorption capacity and has been studied as a contaminant treatment material. Ma found that Cu ions in sewage had a good affinity with red mud, Kocabaş found that 1 g/L ferric ions loaded with red mud could reduce arsenic concentration from 0.3 mg/L to 10 µg/L in pH 7 solution, and Cho mixed red mud with lignin to produce sewage treatment agent with good performance [21,22,23]. Nadaroglu found that red mud was rich in iron oxide, active functional groups which made red mud a solidification stabilizer for heavy metals [24]. Many studies have shown that red mud can improve soil pH value, promote heavy metal ions to form hydroxide precipitation, degrade the exchangeable heavy metals, reduce the mobility and availability of heavy metal ions in the soil, and control heavy metal contaminated soil [25,26,27,28]. Some studies have shown that red mud can be used as a cement stabilizer and weak foundation soil stabilizer because of the flocculating effect of silica, iron oxide and calcium oxide in its components [29,30].

Heavy metal contamination in urban is generally from industrial zone. Heavy metal ions from surface leakage industrial sewage and surface seepage seep into the soil. After that, heavy metal ions gradually diffuse to the deeper soil layer under the effect of water migration, such as rainwater seepage, and finally diffuse into the groundwater. Heavy metal ions migrate with the movement of groundwater, thus making heavy metal ions diffuse to the surrounding soil and groundwater. As heavy metal ions are usually deposited under the foundation of the industrial zone, it is difficult to remove them once they diffuse. In order to prevent the diffusion of heavy metal ions caused by industrial sewage seepage, a hybrid-filling material with high adsorption capacity can be laid on the subsurface to prevent the diffusion of heavy metal sewage seepage (Figure 1). The hybrid-filling materials should not only have the adsorption effect of heavy metal ions but also ensure that their mechanical strength meets the engineering requirements. According to the adsorption and engineering characteristics of red mud, this study creatively proposes to use the mixture of red mud-amended soil as the hybrid-filling material to prevent the diffusion of heavy metal contamination in the industrial zone. On the other hand, this study proposed two different red mud mixing methods, full particle size mixing and partial particle size replacement. The adsorption effect of hybrid material on zinc and lead ions under different mixing ratios was studied, and the desorption characteristics of hybrid-fill material in distilled water and acidic solution were verified. At the same time, the strength and water permeability of the hybrid-fill materials were also measured. This study will provide evidence for using red mud as a heavy metal adsorbent in soil and a new idea for the engineering application of red mud.

## 2. Materials and Methods

### 2.1. Materials

The soil was taken from the construction site in Changchun, which was silty clay with yellow-brown color. The air-dried soil passed through a 1 mm sieve. After sieving, the soil was dried in an oven at 105 °C for 24 h. The red mud was taken from an aluminum plant in Shanxi, which was dark red. The heavy metal content of red mud filtrate is far lower than the standard value of the Identification Standard for Hazardous Waste Extraction Toxicity Identification (GB 5085.3-2007) [31].

The chemical compositions of red mud and field soil were determined by the X-ray fluorescence spectrometer (XRF). The basic physical properties and chemical components of red mud and field soil are shown in Table 1 and Table 2.

The cumulative grain-size distribution curve of red mud and field soil is shown in Figure 2. The heavy metal solutions were prepared by dissolving Zn(NO_3_)_2_·6H_2_O and Pb(NO_3_)_2_ in distilled water. The analytical grade heavy metal nitrates were produced by the Junsei Chemical Company (Tokyo, Japan).

### 2.2. Methods

#### 2.2.1. Soil and Red Mud Mixing Method

In order to get the best result and the optimum mixing ratio, two methods were used in this paper to prepare adsorptive hybrid-fill materials. One method was to add red mud to soil samples in five proportions: 5%, 10%, 15%, 20%, and 25%. The mixed soil samples were named A5, A10, A15, A20, and A25. Another method was to use red mud to partially replace the soil. The study used red mud passing through a #100 sieve (0.149 mm) to partially replace soil whose particle size was the same, with replacement ratios of 10%, 15%, and 20%, respectively. After replacement, they were thoroughly mixed (Figure 3) and named S10, S15, and S20, respectively.

#### 2.2.2. Engineering Characteristics Test

In this study, the strength properties of red mud-amended soil hybrid-fill material were evaluated by direct shear test. The hybrid-fill material was compacted by 95% compaction degree under optimum moisture content and maximum dry density conditions. The test method was performed according to ASTMD3080 [32].

Hydraulic conductivity is a key factor affecting the adsorption of hybrid-fill material. The hydraulic conductivity can affect the flow rate of heavy metal permeate and the contact time between hybrid-fill material particles. As the permeability coefficient of 95% compacted hybrid-fill material was relatively low, the permeability coefficient was measured by the variable head permeability test according to ASTMD 5084 [33].

#### 2.2.3. Adsorption and Desorption Batch Test

The adsorption capacity of red mud-amended hybrid-fill material to zinc and lead ions was first carried out using the batch test. An adsorption test was performed using 2 g of hybrid-fill material with 45 mL of heavy metal solution. Initial concentrations of zinc and lead solutions were 50, 100, 150, 300, 450, 600, 750, and 1000 mg/L, and the final pH of each sample was around 6.5. The mixtures and heavy metal solution were put into the tube for 24 h and in a rotary shaker at room temperature (24 °C). Then the solution was filtered, and each sample’s zinc and lead ions concentration of each sample was measured by Inductively Coupled Plasma Mass Spectrometry(ICP-MS 7900, Agilent, Santa Clara, CA, USA). The Langmuir and Langmuir adsorption isotherm were performed to evaluate the adsorption capacities of hybrid-fill materials.

After the adsorption test, a desorption test was carried out to investigate released heavy metal ions by distilled water and acid solution (pH = 4.5) to prevent secondary contamination by adsorbed heavy metal ions. The hybrid-fill material, after the adsorption test, was dried at 40 °C for 48 h, then put the mixtures into the tube, added 45 mL of distilled water or acid solution was put into the tube. The tubes were shaken for 24 h, and the concentration of each heavy metal was measured by ICP-MS.

#### 2.2.4. Column Test

Column test was designed to simulate the water flow condition in the field; the inflow solution was injected from the top to simulate the contaminant transport in hybrid-fill material. The soil column reflects the unsaturated layer formed by filling above the groundwater level. The column was manufactured in three sizes (20 cm, 25 cm, and 30 cm) in height and 5 cm in diameter. The column was composed of a 2.5 cm sand layer on each side to uniform the flow solution and prevent the loss of mixture in pace with the leakage. Referring to the rainfall intensity of Changchun in the rainy season, the velocity in the soil column was set at 1.5 mL/min. All the mixtures with different mixing ratios were compacted with 95% compactness, and the column test equipment is shown in Figure 4. Three different initial concentrations of heavy metal solutions were 150 mg/L, 300 mg/L, and 450 mg/L. The flow rate of the solution was 1.5 mL/min, and the thickness of the soil column was 20 cm. Furthermore, to examine the adsorption capacity of heavy metal ions on soil columns under different soil layer thicknesses, three soil layer thicknesses of 15 cm, 20 cm, and 25 cm were selected in the experiment. The flow rate and initial concentration of heavy metals were 1.5 mL/min and 300 mg/L. The effluent liquid was collected every 30 min, and the concentration of heavy metal ions was measured by ICP-MS.

## 3. Results and Discussion

### 3.1. Engineering Characteristics Test

Table 3 shows the direct shear test results of red mud-amended hybrid-fill material of different mixing ratios. With the increased red mud content, the internal friction angle of the hybrid-fill material gradually decreased, and the cohesion gradually increased. Under the same mixing ratio, the internal friction angle of the full-size mixed hybrid-fill material was larger than that of the partially replaced mixture (A10 > S10, A15 > S15, A20 > S20). The cohesive force of the full-size mixing method was lower than that of the partial displacement mixing method (A10 < S10, A15 < S15, A20 < S20). This is because red mud has a higher clay content than soil. Therefore, with increased red mud content, the internal friction angle gradually decreases, and the cohesion gradually increases. When the red mud content is less than 20%, the internal friction angle decreases relatively slowly, and the cohesion increases rapidly. However, when the red mud content is more than 20%, the internal friction angle decreases rapidly, and the cohesion growth slows down. Therefore, the mechanical properties are relatively the best when the red mud content is 20%. The internal friction angle of hybrid-fill material was 42.5°–35.2°, and the cohesive force was 25.7 kPa–30.8 kPa, which met the engineering strength requirements of foundation materials.

Hydraulic conductivity is the key factor affecting the adsorption effect of hybrid-fill material for heavy metal ions in industrial zone foundations. Table 3 shows the hydraulic conductivity of hybrid-fill material with different mixing ratios. The hydraulic conductivity of hybrid-fill material ranged from 6.0 × 10^−4^ cm/s to 1.3 × 10^−5^ cm/s, and with the increase of red mud content, the hydraulic conductivity decreased gradually. The column adsorption test verifies that when the hydraulic conductivity is in this range, it can ensure that the hybrid-fill material has enough contact time with contamination and has better removal efficiency for heavy metal ions.

### 3.2. Adsorption Batch Test

Initial concentration is an important factor affecting the adsorption effect of adsorption materials for heavy metal ions. In this study, different initial concentrations of heavy metal solution and different mixing ratios of hybrid-fill material were selected to determine the adsorption capacity of zinc and lead ions by batch test. The adsorption curves of the hybrid-fill material for zinc and lead ions are shown in Figure 5. The results showed that the adsorption capacity of zinc and lead ions increased with the increase of the content of red mud added because the ion exchange capacity of red mud was much stronger than soil particles, so the adsorption capacity of red mud was higher than soil particles. When the content of red mud was 25%, the maximum adsorption capacity of hybrid-fill material for zinc and lead ions reached 6.25 mg/g and 19.87 mg/g, which were 5.07 times and 6.56 times of original soil sample. At the initial stage, the adsorption capacity of heavy metal ions increased rapidly with the increase of initial concentration, then increased slowly and finally remained unchanged. This is because with the initial concentration increase, the solution’s concentration gradient and the driving force increase. The high driving force promotes the diffusion of heavy metal ions in the solution to the adsorbent surface and improves the adsorption efficiency [34,35]. In addition, for a certain amount of adsorption materials, only a limited amount of heavy metal ions can be adsorbed under fixed adsorption sites. With the increase of initial concentration, the number of heavy metal ions adsorbed on the adsorption material gradually increased. When the adsorption sites on the surface of the adsorption material reach saturation, the adsorption capacity of heavy metal ions remains unchanged.

### 3.3. Adsorption Isotherm

In this study, the adsorption mechanism of heavy metal ions on hybrid-fill material and the process of heavy metal ions transfer from an aqueous solution to an adsorbent surface were studied by using an adsorption isotherm. Langmuir adsorption isotherm and Freundlich adsorption isotherm are two commonly used adsorption isotherm models to deeply study the relationship between adsorption capacity and initial concentration. The Langmuir adsorption isotherm represents the monolayer adsorption of heavy metal ions on an adsorbent with a uniform surface, while the Freundlich isotherm represents the non-monolayer adsorption occurring on the heterogeneous surface of the adsorbent [36].

Equation (1) describes the Langmuir isotherm:(1)$$\frac{C_{e}}{Q_{e}}=\frac{1}{ab}+\frac{C_{e}}{a}\;,$$
where a is saturation capacity; b is saturation constant; $C_{e}$ is equilibrium concentration in the aqueous solution; $Q_{e}$ is adsorbed amount.

Equation (2) describes the Freundlich isotherm:(2)$$\ln Q_{e}=\ln K_{F}+\frac{1}{n}\ln C_{e},$$
where $K_{F}$ is Freundlich isotherm constant; $\frac{1}{n}$ is the Freundlich isotherm’s intensity constant; $C_{e}$ and $Q_{e}$ are as previously defined.

Adsorption isotherm model fitting parameters of zinc and lead ion adsorption in hybrid-fill material with different mixing ratios are shown in Table 4 and Table 5. It can be seen from the table that the process of adsorption of zinc ions by hybrid-fill material is well fitted by fitting two isothermal adsorption models, and the R^2^ is greater than 0.78. For the adsorption of zinc ions on a red mud-soil mixture, the R^2^ of the Langmuir adsorption isotherm model (0.94–0.99) is larger than that of the Freundlich adsorption isotherm model (0.78–0.96). For the adsorption of lead ions on hybrid-fill material, Freundlich adsorption isothermal model is more suitable, and the R^2^ (0.89–0.95) is much higher than that of the Langmuir adsorption isotherm model. The results show that the adsorption behavior of zinc ions on hybrid-fill material can be described by Langmuir adsorption isotherm, and the adsorption process of lead ions on hybrid-fill material is more consistent with Freundlich adsorption isotherm. When zinc ions are adsorbed, the adsorbent surface has a uniform surface structure in which all adsorption sites are the same, and the energy is equivalent. Zinc ions form a single adsorption layer on the surface of the adsorption material. Lead ions form complex double-layer adsorption on the surface of the adsorbent, which can also explain why lead ions can be adsorbed more by adsorbent than zinc ions.

### 3.4. Desorption Test

In order to study the stability of hybrid-fill material in different environments after the adsorption of heavy metal ions and whether secondary pollution can be caused by desorption, the distilled water and acidic solution (pH = 4.5) were used to desorb hybrid-fill material after the adsorption of heavy metal ions. Figure 6 and Figure 7 show the desorption rate of zinc and lead ions from hybrid-fill material under distilled water and acid solution, respectively. The results showed that the original soil samples had a higher desorption rate in distilled water and acid solution, even though the adsorption of zinc and lead ions was less in the early stage. The results showed weak adsorption and desorption resistance of heavy metal ions in soil without red mud. After the hybrid-fill material adsorbed zinc and lead ions, the desorption rates of zinc and lead ions in the acid solution were slightly higher than that in distilled water. This is because hydrogen ions can replace heavy metal ions to adsorb on the surface of the adsorbent in the acidic solution, which reduces the adsorption capacity of heavy metal ions by adsorbent material. With the increase of the initial solution concentration, the desorption rate of the two heavy metal ions increased slightly, and the increase of desorption rate mainly occurred in the high concentration stage above 750 mg/L. Under the same conditions, with the increase of red mud content in the hybrid-fill material, the desorption rate decreased gradually, indicating that red mud has a better anti-desorption ability. On the whole, the desorption rates of two kinds of ions adsorbed by hybrid-fill material in distilled water and acidic solution were less than 6% (less than 4% before 750 mg/L), and the desorption rate of lead ions was less than 3%, both of which had stable desorption resistance, and the adsorption for lead ions was more stable than that for zinc ions. The desorption test also verifies the stability of hybrid-fill material for adsorbing heavy metal ions. The test results show that red mud-amended soil has good adsorption capacity and anti-desorption capacity for heavy metal ions and can better prevent the diffusion of heavy metal contamination caused by ground precipitation and permeation.

### 3.5. Column Test

#### 3.5.1. Influence of Initial Concentration

The column leaching test can be used to simulate the adsorption process of hybrid-fill material as the foundation material of an industrial zone to prevent the diffusion of heavy metal contamination under continuous rainfall conditions, which is beneficial to the selection of hybrid-fill material parameters in engineering. Figure 8 shows the breakthrough curve of zinc and lead ions under different initial concentrations at the water inlet of the soil column. Two breakthrough curves in the figure showed similar shapes. For zinc and lead ions, the influence of the initial concentration of feed liquid on the breakthrough curve was consistent. With the increase of the initial concentration of heavy metal ions at the water inlet of the soil column, the breakthrough time was earlier. At the lowest initial concentration of 150 mg/L, the diffusion coefficient and mass transfer coefficient of heavy metal solution decreased due to the relatively low concentration gradient of the heavy metal ions [37]. For zinc ions, the first breakthrough time was 14,760 min (initial concentration was 450 mg/L), and the last breakthrough time was 19,080 min (initial concentration was 150 mg/L); For lead ions, the first breakthrough time was 19,920 min (initial concentration was 450 mg/L), and the last breakthrough time was 24,120 min (initial concentration was 150 mg/L).

The results show that with the Increase of heavy metal concentration at the water inlet of the soil column, the adsorption breakthrough curves of zinc and lead ions gradually become steeper. Under the higher initial concentration of heavy metal conditions, the higher concentration of solution provides a higher concentration gradient, which makes the heavy metal ions move through the solution and adsorbs to the adsorbent material at a higher rate, and makes the adsorption sites of heavy metal ions on the adsorbent material reach saturation faster, so the breakthrough curve becomes steeper. Comparing the breakthrough curve of zinc and lead ions in the figures, it can be seen that the breakthrough time of zinc ions is earlier than that of lead ions at the same initial concentration, which shows that the hybrid-fill material has a good adsorption effect on lead ions. The result is also consistent with the adsorption test results. In addition, with the increase of the concentration of heavy metal solution, the total adsorption capacity of heavy metal ions, the adsorption capacity of equilibrium heavy metals, and the total removal rate of the heavy metals all increase. Other studies also have found similar results [37].

#### 3.5.2. Influence of Hybrid-Fill Material Layer Thickness

Figure 9 shows the breakthrough curves of zinc and lead ions adsorbed in the soil columns under different layer thicknesses of hybrid-fill material. The adsorption capacity of heavy metal ions increases with the increase of the hybrid-fill material thickness. That is because with the increase of the thickness of hybrid-fill material, the mass of adsorption materials of hybrid-fill materials in the soil column also increases, and the increase of adsorption materials will lead to more adsorption sites of heavy metal ions in the soil column, which will lead to the increase of adsorption capacity of heavy metal ions in the soil column [38]. With the increase of hybrid-fill material thickness, the breakthrough time and saturation time obtained from the breakthrough curve tend to increase. That is because more adsorption sites can be provided in the soil column with a thick hybrid-fill material layer. Therefore, in the process of heavy metal ions flowing through the soil column, the adsorbent path of heavy metal ions on hybrid-fill material become longer, and more heavy metal ions can come into contact with hybrid-fill material and be adsorbed by the adsorption materials; thus the breakthrough time and saturation time of the soil column breakthrough curve are prolonged. With the increase of the hybrid-fill material thickness, the breakthrough time of zinc ions changed from 13,320 min to 19,800 min, and the breakthrough time of lead ions changed from 15,120 min to 23,760 min when the heavy metal solution with an initial concentration of 300 mg/L and injection velocity of 1.5 mL/min. Under the same conditions, the breakthrough time of zinc ions was earlier than that of lead ions.

Test results show axial dispersion plays a leading role in the mass transfer process of heavy metal ions adsorption by hybrid-fill material in the soil column. The thickness of hybrid-fill material can affect the axial dispersion, thus affecting the diffusion of heavy metal ions in the hybrid-fill material. The diffusion path of heavy metal ions in the whole soil column in the thinner hybrid-fill material layer is shorter, which reduces the breakthrough time and saturation time of the adsorption process. With the increase of the hybrid-fill material thickness in the soil column, the diffusion path of heavy metal solution in the hybrid-fill material layer is prolonged, which can make heavy metal ions diffuse deeper in the hybrid-fill material layer, and the residence time of heavy metal ions increases, finally the breakthrough time and saturation time of soil column increases. Similar results have been obtained by many scholars [38,39].

#### 3.5.3. Relationship between Soil Column Useful Time and Test Parameters

The thickness of the hybrid-fill material layer and the initial concentration of heavy metal solution are important factors affecting the useful time of the hybrid-fill material soil column. Based on the column test results, the relationship between the useful time of the soil column, the thickness of the hybrid-fill material layer, and the initial concentration of the solution was studied. The useful time of the soil columns obtained under different ratios of outlet concentration to inlet concentration should be different. The study selected the useful time when the ratio of outlet concentration to inlet concentration was 0.5 as the research object. The relationship between the useful time of the soil column, the thickness of the soil column, and the initial concentration of the solution was obtained (Figure 10). It can be seen from Figure 10 that for different zinc and lead solutions, the useful time of the soil column is proportional to the thickness of the hybrid-fill material. With the increase of hybrid-fill material layer thickness, the useful time of the soil column is prolonged. It is inversely proportional to the influent concentration of the solution, and with the increase of influent concentration of the solution, the useful time of the soil column decreases.

The curved surface in Figure 10 can be fitted by the surface equation, and the relationship between the useful time of the soil column, the hybrid-fill material layer thickness, and initial concentration of the solution is obtained. The fitting degree of the fitting formula with fitting points is reasonable (R^2^ > 0.95). The relationship is as follows:

Zinc ions:(3)$$t=-7.1C_{i}+679.7H-71.9\;\left(R^{2}=0.956\right),$$

Lead ions:(4)$$t=-11.09C_{i}+836.9H-13.4\;\left(R^{2}=0.955\right),$$
where t is the useful time of the soil column; H is the hybrid-fill material layer thickness of the soil column, and $C_{i}$ is the initial concentration of the solution.

The useful time of soil column under different hybrid-fill material layer thicknesses and initial solution concentration can be predicted using the fitting formula. In order to explore the feasibility of these fitting formulas, the useful time of soil column measured by experiment and calculated by prediction formula was compared. It can be intuitively seen in Figure 11 that the measured value is very close to the predicted value of the formula. Almost all points in Figure 11 fall on the straight line y=x, which proves that the prediction formula was reasonable.

## 4. Conclusions

This study found that the red mud-amended soil as hybrid-fill material could meet the engineering characteristics and had a higher adsorption capacity of heavy metal ions, which could effectively solve the heavy metal contamination problem caused by surface industrial sewage seepage in the industrial zone.

The adsorption of zinc and lead ions on hybrid-fill materials and engineering characteristics was measured. The red mud content, initial concentration, soil layer thickness, and different mixing methods affected the heavy metal ion removal capacity of the hybrid-fill material. The internal frictional angle and the hydraulic conductivity of red mud-amended soils gradually decreased, and the cohesion gradually increased with the increase of red mud content. The adsorption capacity of heavy metal ions gradually increased, reaching 5.07 times and 6.56 times the adsorption capacity of original soil samples, with the increase of red mud content, while the adsorption capacity of partial particle size displacement mixing method was far greater than that of full particle size displacement mixing method. The desorption rate of two kinds of ions in distilled water and acid solution was less than 6% (less than 4% before 750 mg/L). The adsorption behavior of zinc ions could be described by the Langmuir adsorption isotherm, and the adsorption behavior of lead ions was more consistent with the Freundlich adsorption isotherm. The column test showed that with the increase of the initial concentration of solution and thickness of hybrid-fill material, the breakthrough curve became steeper, and the breakthrough rate became faster. The breakthrough curve of lead ions was steeper than that of zinc ions, proving that hybrid-fill material adsorbed lead ions more effectively than zinc ions. Taking useful time as the dependent variable and solution concentration and soil layer thickness as independent variables, the high fitting formula (R^2^ > 0.95) of zinc and lead ions was obtained, which can effectively evaluate the useful time of hybrid-fill material. In further research, a three-dimensional soil tank test will be used to investigate the in situ performance of hybrid-fill material as foundation materials. The long-term adsorption and desorption capacity, engineering characteristics, and useful time of hybrid-fill material will be evaluated under simulated field conditions.

## Figures and Tables

**Figure 1 ijerph-19-15043-f001:**
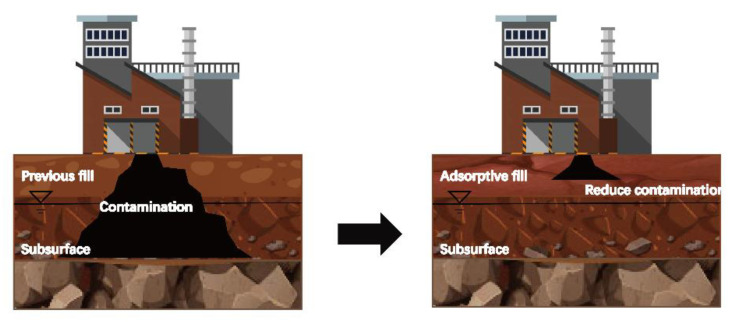
Schematic diagram of heavy metal contamination control in factories.

**Figure 2 ijerph-19-15043-f002:**
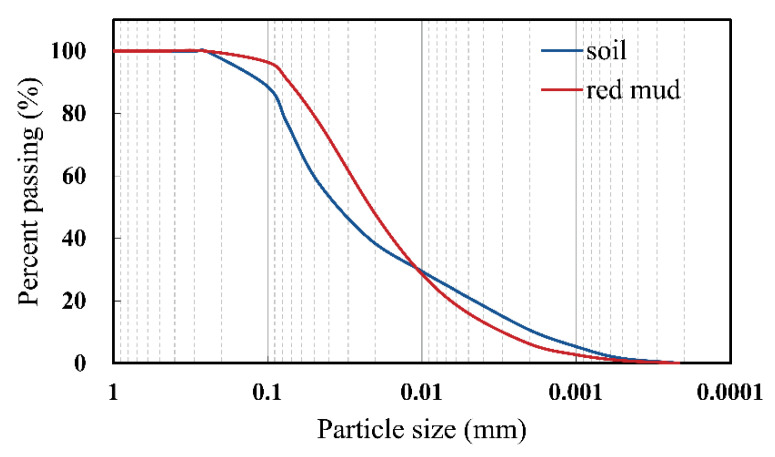
Cumulative grain-size distribution curve of red mud and field soil.

**Figure 3 ijerph-19-15043-f003:**
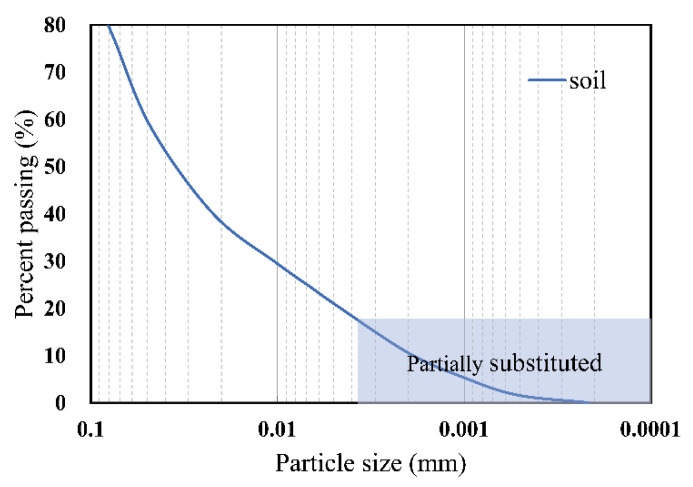
Partial particle displacement method.

**Figure 4 ijerph-19-15043-f004:**
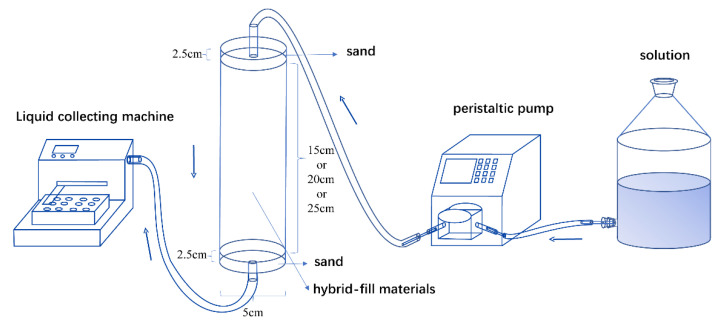
Schematic diagram of the experimental set-up.

**Figure 5 ijerph-19-15043-f005:**
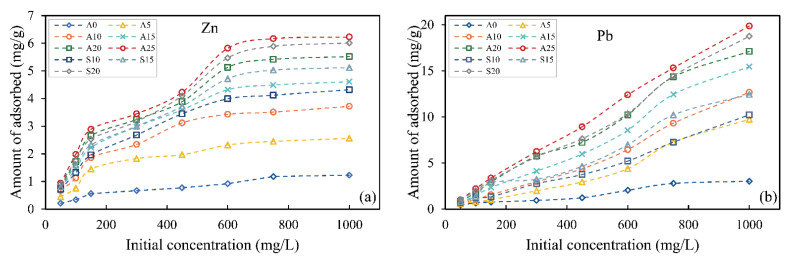
Batch test result of hybrid-fill material adsorbed of zinc ions (**a**) and lead ions (**b**).

**Figure 6 ijerph-19-15043-f006:**
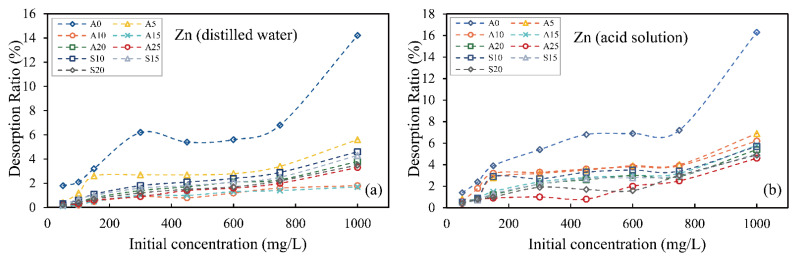
Desorption test result of zinc ions in distilled water (**a**) and acid solution (**b**).

**Figure 7 ijerph-19-15043-f007:**
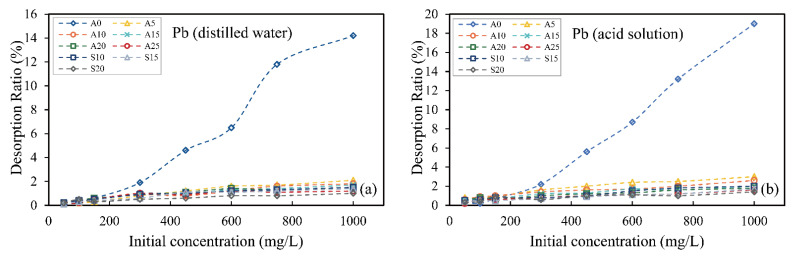
Desorption test result of lead ions in distilled water (**a**) and acid solution (**b**).

**Figure 8 ijerph-19-15043-f008:**
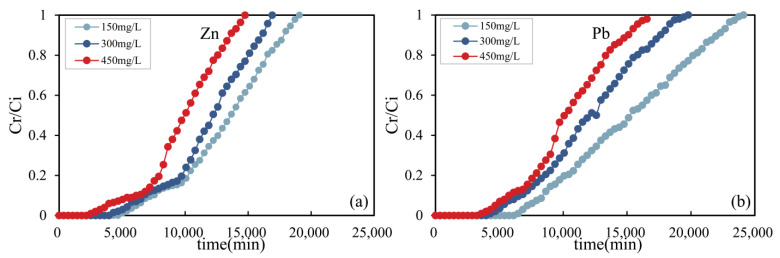
Breakthrough curves of zinc ions (**a**) and lead ions (**b**) at different initial solution concentrations.

**Figure 9 ijerph-19-15043-f009:**
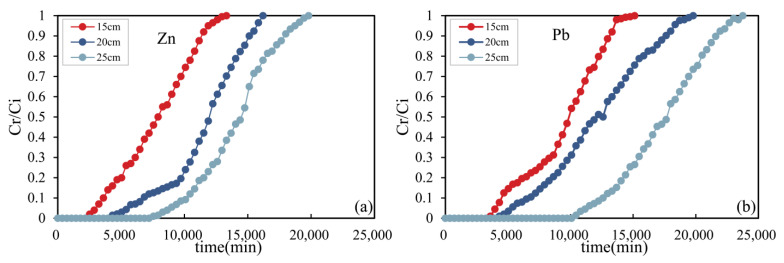
Breakthrough curves of zinc ions (**a**) and lead ions (**b**) under different hybrid-fill material layer thickness.

**Figure 10 ijerph-19-15043-f010:**
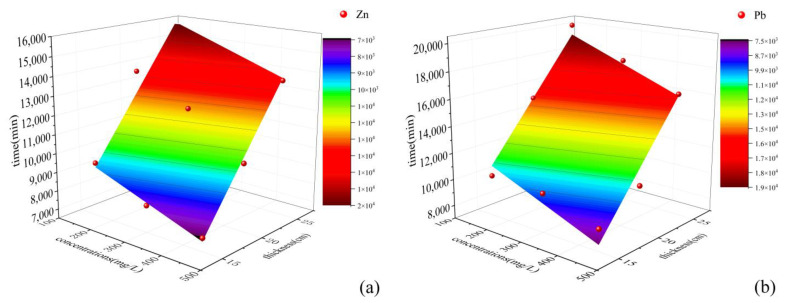
Relationship between the adsorption time of zinc ions (**a**) and lead ions (**b**), the hybrid-fill material layer thickness and the initial concentration of solution.

**Figure 11 ijerph-19-15043-f011:**
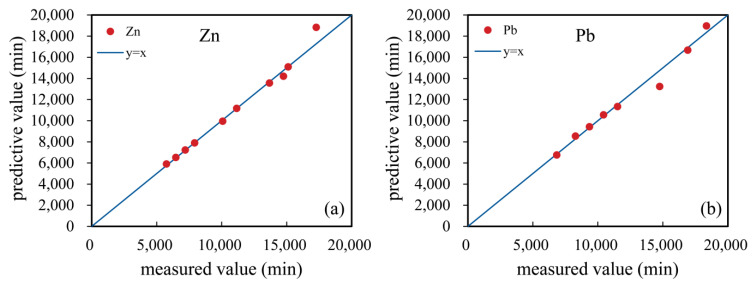
Comparison of the predicted and measured useful times of zinc ions (**a**) and lead ions (**b**).

**Table 1 ijerph-19-15043-t001:** Physical properties of red mud and field soil.

Material	Field Soil	Red Mud
optimum moisture content (%)	17.80	72.50
specific gravity	2.72	2.68
pH	6.84	10.38
cation exchange capacity (meq/100 g)	9.45	56.81
specific surface area (m^2^/g)	11.97	78.09
liquid limit (%)	37.60	74.60
plastic limit (%)	17.30	47.20

**Table 2 ijerph-19-15043-t002:** Chemical components of red mud and field soil.

Formula	Field Soil	Red Mud
SiO_2_	59.50%	40.96%
Al_2_O_3_	19.28%	22.13%
Na_2_O	1.32%	0.07%
Fe_2_O_3_	6.68%	3.38%
CaO	0.74%	22.96%
TiO_2_	-	2.09%
K_2_O	2.96%	-
LOI	9.46%	2.41%

**Table 3 ijerph-19-15043-t003:** Hydraulic conductivity and shear strength parameters of hybrid-fill material.

Sample	Hydraulic Conductivity (cm/s)	Internal Friction Angle (°)	Cohesion (kPa)
A0	6.0 × 10^−4^	43.2	23.9
A5	2.8 × 10^−4^	42.5	25.7
A10	1.0 × 10^−4^	40.9	26.4
S10	8.4 × 10^−5^	40.1	26.8
A15	6.5 × 10^−5^	39.8	27.4
S15	4.1 × 10^−5^	39.3	28.0
A20	3.0 × 10^−5^	38.9	30.3
S20	2.2 × 10^−5^	36.8	30.6
A25	1.3 × 10^−5^	35.2	30.8

**Table 4 ijerph-19-15043-t004:** Langmuir and Freundlich model parameters of zinc ions adsorption by hybrid-fill material.

Zinc	Langmuir Isotherm Constants			Freundlich Isotherm Constants		
	1/a	1/ab	R^2^	K_F_	1/n	R^2^
A0	0.3181	98.7890	0.9446	0.052707	0.5813	0.9609
A5	0.1659	25.8330	0.9888	0.239046	0.4707	0.8658
A10	0.1178	14.0480	0.9942	0.470904	0.4213	0.8787
A15	0.0972	8.9995	0.9945	0.722672	0.3897	0.8567
A20	0.0826	6.8458	0.9783	1.169177	0.3538	0.8002
A25	0.0750	4.9881	0.9654	1.560178	0.3083	0.7819
S10	0.1015	11.6230	0.9960	0.547825	0.4213	0.8856
S15	0.0870	8.6876	0.9842	0.784664	0.3899	0.8571
S20	0.0752	6.9333	0.9653	1.059821	0.3636	0.8762

**Table 5 ijerph-19-15043-t005:** Langmuir and Freundlich model parameters of lead ions adsorption by hybrid-fill material.

Lead	Langmuir Isotherm Constants			Freundlich Isotherm Constants		
	1/a	1/ab	R^2^	K_F_	1/n	R^2^
A0	0.1112	65.2630	0.5196	0.109285	0.5705	0.9074
A5	0.0057	42.1690	0.0075	0.043630	0.8993	0.8972
A10	−0.0095	25.2280	0.0844	0.044396	0.9978	0.9572
A15	−0.0019	12.0130	0.0068	0.084568	1.0060	0.9584
A20	0.0111	4.8554	0.2041	0.329131	0.8890	0.9205
A25	0.0187	1.1193	0.4756	4.766918	0.3550	0.5556
S10	0.0213	22.9740	0.2149	0.134472	0.7467	0.9370
S15	0.0261	11.4780	0.3012	0.360667	0.6495	0.8071
S20	0.0248	2.2065	0.5767	0.982063	0.6584	0.9391

## Data Availability

Not applicable.

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
