# Peer review of "Red Mud-Amended Soil as Highly Adsorptive Hybrid-Fill Materials for Controlling Heavy Metal Sewage Seepage in Industrial Zone"

_ijerph, 2022, doi:10.3390/ijerph192215043_

Round 1
Reviewer 1 Report
1、There are 7 keywords that need to be selected.
2、The introduction can also be summarized and improved.
3、The effective data in Table 1, Table 2, Table 5 and Table 6 are not uniform.
4、What does "-" in Table 2 mean?
5、The connecting pipe in Figure 4 is not beautiful, so it is recommended to redraw it.
6、Tables 3 and 4 are proposed to be merged.
7、The sentence "Heavy metals can cause irreversible damage to the environment," in line 35 is not completely correct. Heavy metal contaminated soil can be repaired.
8、The sentence“and the leakage of industrial sewage seep into the industrial complex foundation soil,”in line 43,How to understand?
9、The fitting degree in Figure 11 is only 0.95, and one point deviates a lot. Why “The fitting degree of fitting formula with fitting points is higher (R2 > 0.95). ”(in line 384)?
Author Response
Thank you very much for your letter and comments about our manuscript. Base on the comments, the manuscript has been carefully revised. We submit a cover letter to elaborate on revised part. Please see the attachment.

Reviewer 2 Report
The article entitled "Red mud-amended soil as highly adsorptive hybrid-fill materials for controlling heavy metal sewage seepage in industrial zone" was very interesting and very useful for the reduction of environmental pollution in industrial areas.
However, the manuscript can be improved if the authors will do the following corrections:
1. In the Introduction section must be inserted few sentences which will describe the originality/novelty of the paper.
2. The measurement unit ug/L must be replaced with µg/L.
3. In Table 1 the column with Material, all materials must be written using lowercase or uppercase, not mixed.
4. In Table 2 the author must insert the measurement unit(s) for the chemical components and must describe the analytical technique used for these determinations.
5. In the Sub-section 2.2.3. - after the sentence "Then the 149 solution was filtered, zinc and lead ions concentration of each sample was measured by 150 ICP-MS (Inductively Coupled Plasma Mass Spectrometry)." must provide the data about quality control of measurements, as well as the parameters and method of detection (STD/KED).
6. In the references list, the reference 34 must be revised.
Author Response

(The authors gave the same response as above.)

Reviewer 3 Report
1) The parameters taken for analyzing the adsorption capacities of red-mud amended soils like shear strength permeability test, adsorption and desorption batch test, and column test, signify the soil characteristics. How can these parameters be evaluated for estimating the adsorption capacities of lead and zinc ions and engineering characteristics on red mud-amended soils?
2) The optimum content of red mud in hybrid-fill material can be determined as 20%. How is this justified in the manuscript?
3) What challenges and future prospects can refrain red mud for amendment of heavy metal contaminated soil? Thus, the manuscript can be accepted after minor revision of the above mentioned comments.
The manuscript can be improved for experimental design and proper observations to analyze the results. Also, English grammar should be checked and repetition be avoided.
Author Response

(The authors gave the same response as above.)

Reviewer 4 Report
Dear author,
Congratulations on a scientifically sound and relevant study. It is important to follow this study up as mentioned in your article: "The long-term adsorption and desorption capacity, engineering characteristics and useful time of hybrid-fill material will be evaluated under simulated field conditions".
Please see my comments in the attached article -please get the article proofread for grammar and English language, before finalization.

Author Response

(The authors gave the same response as above.)
